# *HER2*-Positive Breast Cancer Patients with Pre-Treatment Axillary Involvement or Postmenopausal Status Benefit from Neoadjuvant Rather than Adjuvant Chemotherapy Plus Trastuzumab Regimens

**DOI:** 10.3390/cancers13030370

**Published:** 2021-01-20

**Authors:** Enora Laas, Arnaud Bresset, Jean-Guillaume Féron, Claire Le Gal, Lauren Darrigues, Florence Coussy, Beatriz Grandal, Lucie Laot, Jean-Yves Pierga, Fabien Reyal, Anne-Sophie Hamy

**Affiliations:** 1Department of Surgery, Institut Curie, 5 rue d’Ulm, 75005 Paris, France; enora.laas@curie.fr (E.L.); jeanguillaume.feron@curie.fr (J.-G.F.); claire.legal@curie.fr (C.L.G.); lucie.laot@aphp.fr (L.L.); 2Gynecology Department, Beaujon Hospital, 92210 Clichy, France; arnaud.bresset@aphp.fr; 3Residual Tumor & Response to Treatment Laboratory, RT2Lab, Translational Research Department, INSERM, U932 Immunity and Cancer, Institut Curie, 75005 Paris, France; lauren.darrigues@curie.fr (L.D.); beatriz.grandalrejo@curie.fr (B.G.); anne-sophie.hamy-petit@curie.fr (A.-S.H.); 4Department of Medical Oncology, Institut Curie, 75005 Paris, France; florence.coussy@curie.fr (F.C.); jean-yves.pierga@curie.fr (J.-Y.P.)

**Keywords:** breast cancer, neoadjuvant chemotherapy, adjuvant chemotherapy, *HER2*-positive tumors

## Abstract

**Simple Summary:**

Neoadjuvant chemotherapy strategy (NAC) is a standard of care for Human Epidermal Growth Factor Receptor-2 (*HER2*)-positive early breast cancer but there is no proven beneficial evidence in terms of survival compared to the adjuvant chemotherapy strategy. Our retrospective study found a survival benefit in NAC strategy particularly in clinical Nodepositive and postmenopausal patients.

**Abstract:**

Background: No survival benefit has yet been demonstrated for neoadjuvant chemotherapy (NAC) against *HER2*-positive tumors in patients with early breast cancer (BC). The objective of this study was to compare the prognosis of *HER2*-positive BC patients treated with NAC to that of patients treated with adjuvant chemotherapy (AC). Materials and methods: We retrospectively analyzed disease-free (DFS) and overall survival (OS) in 202 *HER2*-positive patients treated with NAC and 701 patients treated with AC. All patients received trastuzumab in addition to chemotherapy. Patient data were weighted by a propensity score to overcome selection bias. Results: After inverse probability of treatment weights (IPTW) adjustment, no difference in DFS (*p* = 0.3) was found between treatments for the total population. However, after multivariate analysis, an interaction was found between cN status and chemotherapy strategy (IPTW-corrected corrected Hazard ratio cHR = 0.52, 95% CI (0.3–0.9), *p*_interaction_ = 0.08) and between menopausal status and chemotherapy (CT) strategy (cHR = 0.35, 95%CI (0.18–0.7)) *p*_interaction_ < 0.01). NAC was more beneficial than AC strategy in cN-positive patients and in postmenopausal patients. Moreover, after IPTW adjustment, the multivariate analysis showed that the neoadjuvant strategy conferred a significant OS benefit (cHR = 0.09, 95%CI [0.02–0.35], *p* < 0.001). Conclusion: In patients with *HER2*-positive BC, the NAC strategy is more beneficial than the AC strategy, particularly in cN-positive and postmenopausal patients. NAC should be used as a first-line treatment for *HER2*-positive tumors.

## 1. Introduction

Epidermal Growth Factor Receptor-2 (*HER2*)-positive breast cancers (BCs) are carcinomas characterized by the amplification and overexpression of the *HER2* tyrosine kinase receptor gene (17q12), and they account for 15–20% of all BCs [1]. Before the advent of trastuzumab and anti-HER2-targeted therapies (i.e., lapatinib, pertuzumab and TDM-1), the rates of early distant metastatic events in these patients were high. Since the introduction of trastuzumab into routine use in the adjuvant setting in the middle of the first decade of this century, the combination of trastuzumab and chemotherapy has considerably improved the prognosis of patients with *HER2*-positive early BC [2].

Neoadjuvant chemotherapy (NAC) is now widely used in patients with early BC, to achieve breast-conserving surgery (BCS) through prior tumor size reduction, and it is now a standard of care for patients with *HER2*-positive BC [3,4,5,6,7]. In addition to BCS, NAC provides the possibility of in-vivo chemosensitivity tests of tumor burden and an opportunity to observe clinical and pathological responses in each patient. NAC is therefore an effective and informative strategy for testing novel therapies, providing an opportunity to adapt adjuvant strategy (TDM-1) [8,9].

No difference in overall survival (OS) has been detected to date between neoadjuvant and adjuvant strategies, neither in the total population of BC patients nor in the population of patients with *HER2*-positive tumors. Indeed, in pivotal trials addressing the equivalence between these two strategies, *HER2* status was not routinely evaluated [10,11,12]. In an Early Breast Cancer Trialists’ Collaborative Group EBCTCG meta-analysis pooling individual data from 4756 patients included in randomized trials of NAC vs. AC strategies (n = 10), no significant difference between these strategies was found for distant recurrence at 15 years (rate ratio 1.02 (95% CI 0.92–1.14); *p* = 0.66), BC mortality (1.06 (0.95–1.18); *p* = 0.31), or death from any cause (1.04 (0.94–1.15); *p* = 0.45), regardless of histological subtype [13]. 

The aim of this study was to compare the impact of NAC and AC strategies (in addition to trastuzumab in both groups) on survival outcomes, in *HER2*-positive BC patients, using a propensity score to avoid selection bias.

## 2. Materials and Methods

### 2.1. Patient Selection

Patients with T1-3N0-3M0 invasive *HER2*-positive tumors undergoing primary treatment with a combination of NAC and trastuzumab at Institut Curie between 1 January 2005 and 31 December 2012 (*n* = 202) (NEOREP cohort CNIL declaration number 1547270) were selected to constitute the NAC group. These patients were compared with 701 patients with T1-3N0-3M0 invasive *HER2*-positive breast tumors treated with a combination of AC and *HER2*-targeted therapies between 1 January 2005 and 31 December 2012, identified from the Institut Curie prospective breast cancer database. Unifocal, unilateral, non-recurrent, non-metastatic tumors were included. This study was approved by the Breast Cancer Study Group of Institut Curie and was conducted according to institutional and ethical rules concerning research on tissue specimens and patients. Informed consent from patients was not required. Information concerning clinical (age, menopausal status, body mass index, clinical tumor size, clinical nodal status) and tumor characteristics (histological size in the AC group and histological response to NAC, lymph node involvement, mitotic index, ki67, histological tumor grade, estrogen receptor (ER) status, progesterone receptor (PR) status, *HER2* status) was retrieved from patients’ medical records. Clinical size and clinical nodal status were used for the analysis, as they play a crucial role in decisions regarding tumor management.

### 2.2. Tumor Samples

Histological grade was determined according to the Elston-Ellis (EE) modification of the Scarff-Bloom-Richardson grading system [14] on post-operative tumor samples for the AC group and needle core biopsy specimens for the NAC group. Hormone receptor (HR) expression was analyzed by immunohistochemistry. Tumors were considered positive for ER or PR if 10% of the carcinoma cells displayed positive staining, as recommended by French guidelines [15]. Tumors were considered to be hormone receptor (HR)-positive if they were positive for either ER or PR, and HR-negative if they were negative for both ER and PR. *HER2* status was determined according to American Society of Clinical Oncology (ASCO) recommendations [16]. Scores of 3+ were considered positive, and scores of 1+/0 were considered negative. Tumors with scores of 2+ were further tested by Fluorescence In Situ Hybridization (FISH). We assessed *HER2* gene amplification by calculating the mean *HER2* signal per nucleus for a mean of 40 tumor cells per sample. A *HER2*/CEN17 ratio ≥ 2 was considered positive and a ratio < 2 negative.

### 2.3. Treatment Protocol

Patients were treated according to French national guidelines. The decision to administer NAC was made at a multidisciplinary medical meeting, in cases for which conservative surgery was not possible or for patients with high-risk factors (positive nodal status, high EE grade, high mitotic index, etc.). Chemotherapy regimens changed over time. In the adjuvant setting, chemotherapy was initiated four to eight weeks after surgery. The chemotherapy regimen was anthracycline-based in 101 patients (14%), taxane-based in 12 (2%), and a sequential anthracycline–taxane regimen (three cycles of epirubicin-cyclophosphamide followed by three cycles of docetaxel, or 12 cycles of paclitaxel weekly) was administered in 546 (78%) patients. Trastuzumab was administered at a dose of 8 mg/kg (loading dose) and then 6 mg/kg every 3 weeks. Weekly trastuzumab treatment was then continued for one year.

In the neoadjuvant setting, three patients (1%) had an anthracycline-based regimen, 16 (8%) had a taxane-based regimen, and 175 (87%) had a sequential anthracycline–taxane regimen (four cycles of epirubicin-cyclophosphamide followed by four cycles of docetaxel or 12 cycles of paclitaxel weekly). Trastuzumab was administered at a dose of 8 mg/kg (loading dose) and then 6 mg/kg every 3 weeks. 

None of the patients received pertuzumab, since it was off-label in France during the period of the study, in the non-metastatic setting.

Surgery was performed three to six weeks after the last chemotherapy session. Weekly trastuzumab was then continued for one year after surgery. 

In the adjuvant setting, sentinel lymph node biopsy (SLND) was performed in case of cN0 and axillary lymph node dissection (ALND) for patients who were cN-positive. In the neoadjuvant setting, during the study period, French and European recommendations stated for systematic ALND without SLND after NAC, so most patients underwent ALND [17,18].

Pathological complete response (pCR) was defined as no residual invasive/non-invasive cancer in the breast and nodes (ypT0-ypN0).

Patients received adjuvant radiotherapy according to national guidelines. The indications for radiotherapy were breast-conserving surgery, radical mastectomy in cases of initial T3 tumors, patients with axillary lymph node involvement, and patients with high-risk node-negative breast cancer. Adjuvant endocrine therapy (tamoxifen, aromatase inhibitor, or Gonadotropin-Releasing Hormone agonists) was prescribed when indicated to patients displaying hormonal receptor expression after radiotherapy. After completing treatment, patients were followed every four months for the first two years, every six months for the next three years, and then annually after five years. Follow-up consisted of a clinical examination associated with mammography and mammary ultrasound once per year.

### 2.4. Study Endpoint

The aim of the study was to analyze and compare prognosis and oncological outcomes between NAC and AC strategies. Disease-free survival (DFS) was defined as the time from surgery to locoregional, distant recurrence or death, whichever occurred first. Overall survival (OS) was defined as the time from surgery to death. Patients for whom none of these events were recorded were censored at the date of last known contact. 

Distant recurrence-free survival (DRFS) was defined as the time from surgery to distant recurrence or death, whichever occurred first. 

### 2.5. Statistical Analysis

Qualitative variables were compared with Chi-squared or Fisher’s exact tests, and quantitative variables were compared with Student’s *t*-tests. A significance threshold of 5% was applied. As patients were not randomly allocated to the AC or NAC arms, a propensity score (PS) [19,20,21,22] was calculated to control for selection bias. The characteristics of the patients in the two groups at baseline were compared in univariate analysis. A multivariate logistic regression model was then used to generate the PS. The covariates included in the model were patient clinical characteristics (age, menopausal status, body mass index, initial clinical tumor size), histological characteristics at diagnosis (Elston-Ellis grade), and initial clinical nodal status. We created balanced cohorts, using the inverse probability of treatment weights (IPTW), defined as the inverse of the propensity score for patients in the NAC population and 1/(1 − *propensity score*) for patients in the AC population. The IPTW adjustment method is a valuable method to obtain unbiased estimates of average treatment effects [23,24] and has already been used in breast cancer studies [25,26]. Weight outliers lying above the 99th percentile were truncated to prevent sparse data. Missing data were handled by multiple imputation with chained equations. The benefit of NAC was estimated in the initial and IPTW-corrected populations, from Kaplan–Meier curves. Survival was compared between groups by performing log-rank tests on the initial population and a weighted univariate Cox model in the IPTW-corrected population [27]. An IPTW-corrected multivariate Cox model analysis was then performed, to take into account potential confounding factors not included in the PS (mostly surgical management). Hazard ratios for relapse, corrected by the IPTW procedure, are referred to as corrected HR (cHR) hereafter. We tested the hypothesis of potentially different effects of NAC in different subgroups, by including pairwise interaction terms in the Cox model. Due to the lack of statistical power for analyzing interactions [28], a *p*-value of 0.10 or lower was considered statistically significant. Analyses were performed with R software, version 3.3 [29].

## 3. Results

Overall, 903 patients with *HER2*-positive BC underwent NAC (*n* = 202) or AC (*n* = 701) with *HER2*-trastuzumab (Table 1). Before IPTW correction, patients from the NAC group were younger than those from the AC group (48.2 vs. 53.6 years old; *p <* 0.0001), had tumors of a larger median size (40 vs. 20 mm; *p <* 0.001), were more likely to have clinical nodal involvement (60.1% vs. 18.3%, *p <* 0.0001), and to be treated by BCS (70.4% vs. 40.9%; *p <* 0.0001). Hormone receptor status did not differ between groups (55% of patients in each group had HR-positive tumors, *p = 0.9*), and was not balanced between pre- and post-menopausal patients (60% HR-positive for premenopausal patients vs. 50% for post-menopausal patients, *p* = 0.03, Figure 1). The pathological complete response (pCR) rate was 38.5% in the NAC group.

The median time from BC diagnosis to treatment was significantly lower in the NAC group than in the AC group (time from BC diagnosis to first chemotherapy: 21 days vs. 83 days, *p* < 0.001, Figure A1A; time from BC diagnosis and first trastuzumab injection: 103 days vs. 156 days, Figure A1B). With a median follow-up of 39.2 months for the NAC group and 48.7 months for the AC group, there were 74 events (NAC: *n* = 20; AC: *n* = 54) and 20 deaths (NAC: *n* = 2; AC: *n* = 18). DFS and OS did not differ significantly between the groups before IPTW correction (Figure A1E,F).

### 3.1. Survival Analysis after IPTW Correction

The initial distribution of propensity scores was not well balanced between the NAC and AC groups (Figure A1C). Figure A1D in the Appendix A shows the standardized mean difference between the two groups for each variable before and after IPTW correction. After IPTW correction (represented by the red dots), all the variables were comparable between the NAC and AC groups, except for mastectomy rate and clinical tumor size, which remained higher in the NAC group. 

### 3.2. Disease-Free Survival

After univariate IPTW-corrected analysis, the NAC and AC strategies were not significantly associated with DFS (cHR = 0.89, 95%CI: (0.61–1.3), *p* = 0.3) (Figure 2A). When analyzing the association between CT strategy and DFS according to clinical and pathological factors, the NAC strategy was found to be significantly associated with DFS in the group of patients with baseline clinical nodal involvement (cN-positive) (*p_i_*_nteraction_ = 0.04) and in premenopausal patients (*p*_interaction_ < 0.01) (Figure 3A). 

After multivariate analysis and adjustment for surgical management, significant interactions remained between cN status and CT strategy, and between menopausal status and CT strategy (*p*_interaction_ = 0.08 and <0.01, respectively), indicating that the NAC strategy was more beneficial than the AC strategy in cN-positive patients (cHR = 0.52, 95% CI (0.3–0.9)) and in postmenopausal women (cHR = 0.35, 95%CI (0.18–0.7)) (Figure 3B). No other factor analyzed in this study was significantly associated with differences regarding the impact of the CT strategy.

Similar results were found regarding the DRFS: the NAC and AC strategies were not significantly associated with DRFS (Figure A2, while significant interactions existed between cN status and CT strategy, and between menopausal status and CT strategy (Figure A3A,B).

### 3.3. Overall Survival

Very few deaths (*n* = 2) were observed in the NAC group, limiting comparisons of the impact of the two strategies on overall survival. After IPTW-corrected univariate analysis, the NAC strategy was significantly associated with a greater benefit in terms of OS at five years than the AC strategy (cHR = 0.12, 95% CI (0.03–0.47), *p* = 0.002) (Figure 2B). After multivariate analysis and adjustment for type of surgery and radiotherapy, this survival benefit remained statistically significant (cHR = 0.09, 95%CI (0.02–0.35), *p* < 0.001) (the figure is not shown because the confidence interval was infinite, given the small number of deaths).

## 4. Discussion

In this study, we compared the NAC and AC strategies in *HER2*-positive BC patients. We found that the NAC strategy was significantly associated with longer DFS in patients with a clinical positive nodal status at baseline, and in postmenopausal patients. Moreover, NAC was associated with a significant survival benefit over AC in the whole population. 

This is, to our knowledge, one of the first studies to report an impact of treatment strategy (NAC vs. AC) in a subset of patients: the *HER2*-positive population. In a recent study of Pomponio et al., using IPTW to deal with the selection bias showed no difference between AC and NAC strategy in early *HER2*-postive BC [30]. Many studies have reported higher rates of breast-conserving surgery in patients treated with NAC, but a recent EBCTCG meta-analysis [13] found that NAC was as effective as AC in reducing the risk of distant recurrence and death from BC. NAC was associated with a higher local recurrence rate at 15 years than AC (21.4% vs. 15.9% respectively), but these results must be interpreted with care, as chemotherapy regimens, surgical, and radiation techniques have improved over time. Stankowski-Drengler et al. [31] identified three studies analyzing outcomes, by chemotherapy strategy, and based on receptor subtype. One of these studies reported analyses for the *HER2*-positive subgroup showing an OS benefit of the NAC strategy (HR = 0.1, 95%CI [0.02–0.58], *p* = 0.01). However, the *HER2*-positive subgroup only contained 30 patients, and patients treated with targeted therapies against *HER2* were excluded [32]. 

Little scientific evidence is available concerning the subgroups of patients with *HER2*-positive BC that should be offered a NAC strategy rather than an AC strategy. A large tumor size is considered to be a genuine indication for NAC, as it increases the likelihood of breast-conserving surgery, and the neoadjuvant strategy was validated as a standard of care in the 2019 St Gallen Consensus for stage II–III tumors [33]. The availability of drugs also guides therapeutic strategies. The 2019 National Comprehensive Cancer Network guidelines validated a NAC regimen combining trastuzumab and pertuzumab for *HER2*-positive tumors with a diameter exceeding 2 cm and for node-positive patients [34]. The use of adjuvant drugs, such as TDM-1, in the post-neoadjuvant setting also provides a major argument in favor of NAC. The KATHERINE trial found a benefit of the association of trastuzumab and emtansine (TDM-1) in the adjuvant setting for patients with residual disease after NAC, with a risk of recurrence of invasive BC or death 50% lower in the TDM-1 group (HR = 0.50; 95% CI (0.39–0.64)) [9]. Since then, neoadjuvant strategy tends to be the standard for the *HER2*-positive tumors in most recommendations [6,35].

In our cohort, menopausal status was significantly associated with DFS. These finding are consistent with the retrospective study by Kim et al. on 229 *HER2*-positive tumors treated by NAC plus trastuzumab, which reported significantly lower pCR rates in young and premenopausal patients [36].

In contrast, another study with 475 *HER2*-positive tumors found no difference in pCR rates between two age groups (<40 years vs. ≥40 years) [37], but none of these patients were treated with neoadjuvant trastuzumab. In 2015, a meta-analysis from the German Breast Group (GBG) and the German Gynecological Oncology Working Group-Breast (AGO-B) study group focusing on young patients found no significant difference in pCR rates as a function of age or HR status, in patients with *HER2*-positive tumors [38].

We also found that the NAC strategy was beneficial in the population of patients with positive nodal status at baseline. Axillary node involvement is associated with more advanced tumor stages at BC diagnosis, and is correlated with poorer BC prognosis and higher rates of distant metastasis [39,40,41]. Several hypotheses can be put forward to explain our findings. First, the NAC strategy results in a shorter time-to-first systemic treatment. As *HER2*-positive BCs are highly sensitive to chemotherapy combined with anti-*HER2* treatment, the impact of a delay in initiating trastuzumab treatment and/or chemotherapy might be particularly important in this subpopulation. However, the REMAGUS02 trial randomly assigned *HER2*-positive BC patients to two arms (NAC alone or NAC plus trastuzumab), and reported that there was no survival advantage associated with the early introduction of trastuzumab [42]. Second, there is evidence to suggest that trastuzumab has an immunological mode of action, including the induction of antibody-dependent cellular cytotoxicity, endogenous humoral, and enhanced T cell-mediated immune responses, and higher levels of tumor infiltration with immune effectors, including natural killer cells. It remains unknown whether these mechanisms act differently before or after the surgical removal of the tumor, but it could be argued that such mechanisms could be attenuated with the tumor in place [43,44]. Third, although purely speculative, we can hypothesize that axillary involvement is of importance in the context of this cancer with a large immunological dimension for which antibody-based therapies are available [45]. 

Our study has several limitations. First, as this study was retrospective, patients were not randomly allocated to the NAC and AC strategies. We tried to limit selection bias by weighting according to the probability of treatment allocation with a propensity score. This method has been shown to be an effective alternative to consistently reduced systematic baseline differences [20]. Second, none of the patients received pertuzumab in the neoadjuvant setting. The addition of pertuzumab (a humanized monoclonal antibody targeting an epitope of *HER2* different from the one targeted by trastuzumab) to trastuzumab associated with docetaxel has been shown to yield higher rates of pCR, and a higher five-year DFS, in early, locally advanced and inflammatory BC [46,47]. In 2019, Tolaney et al. published an updated seven-year follow-up for an adjuvant paclitaxel and trastuzumab protocol, demonstrating the efficacy of an adjuvant anthracycline-free regimen for *HER2*-positive, node-negative tumors of 3 cm or less in diameter [48]. With its very low rates of recurrence and highly favorable toxicity profile, this adjuvant strategy has become a standard of care in the management of small node-negative *HER2*-positive BC. It remains unknown whether these results would be reproducible with such a chemotherapy regimen. Finally, we found discrepancies between the results for DFS and OS. These differences could be explained by the higher death rate in the AC group than in the NAC group, and a higher local recurrence rate in the NAC group, possibly due to the higher rate of conservative surgery. However, similar results were found regarding DRFS.

Our study also has several strengths. This is to our knowledge the first study demonstrating an OS benefit of trastuzumab containing neoadjuvant chemotherapy for *HER2*-positive BC patients, with a large statistical power and a long-term follow-up. Moreover, as NAC has progressively become a standard of care over the last years, historical data comparing AC and NAC in these tumors may be difficult to encounter from now on, making these data unique. Finally, beyond new drugs and new predictive markers, this is one of the first studies supporting the hypothesis that therapeutic strategies tailored to each individual may influence the natural history of BC. 

## 5. Conclusions

In conclusion, NAC remains the standard of care for *HER2*-positive BCs [49], and our study provides additional evidence to support the treatment of patients with *HER2*-positive tumors and baseline clinical nodal involvement with NAC rather than AC. Validation studies determining whether similar subgroups of patients derive benefits from the combination of a NAC regimen with anti-*HER2* dual-blockade with trastuzumab and pertuzumab are eagerly awaited.

## Figures and Tables

**Figure 1 cancers-13-00370-f001:**
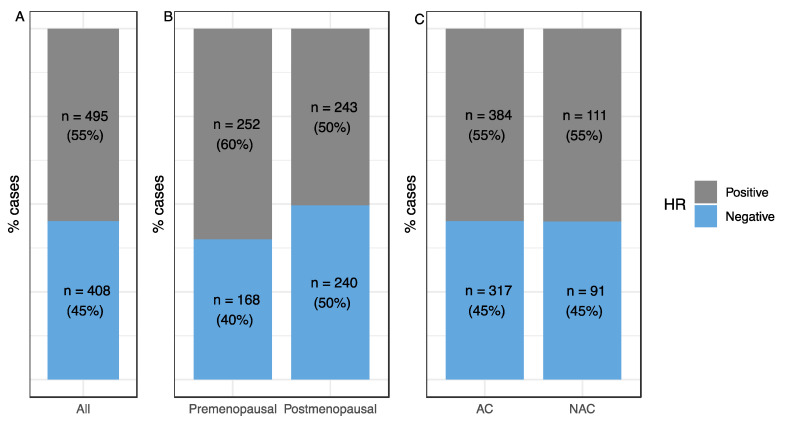
Hormone receptor repartition. (**A**) Whole population, (**B**) according to menopausal status, (**C**) according to chemotherapy strategy. NAC: Neoadjuvant chemotherapy; AC: Adjuvant chemotherapy; HR: Hormone receptor.

**Figure 2 cancers-13-00370-f002:**
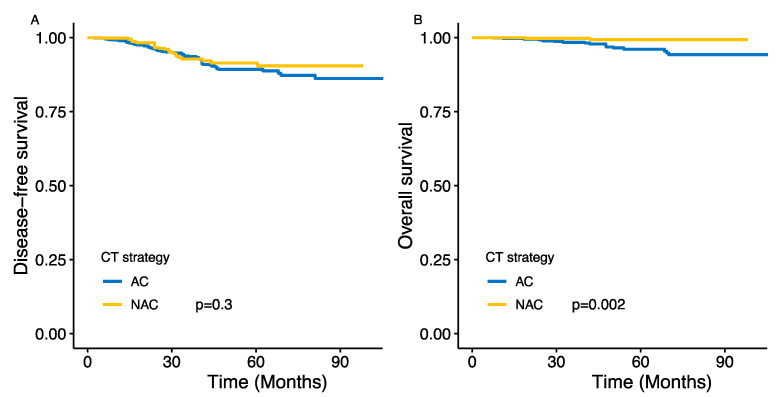
Association between treatment strategy (adjuvant or neoadjuvant) and survival after IPTW correction. (**A**) Disease-free survival; (**B**) Overall survival. CT: chemotherapy; NAC: Neoadjuvant chemotherapy; AC: Adjuvant chemotherapy; IPTW: inverse probability of treatment weights. The *p*-value is calculated with a weighted Cox model. No risk table is provided as the population is weighted).

**Figure 3 cancers-13-00370-f003:**
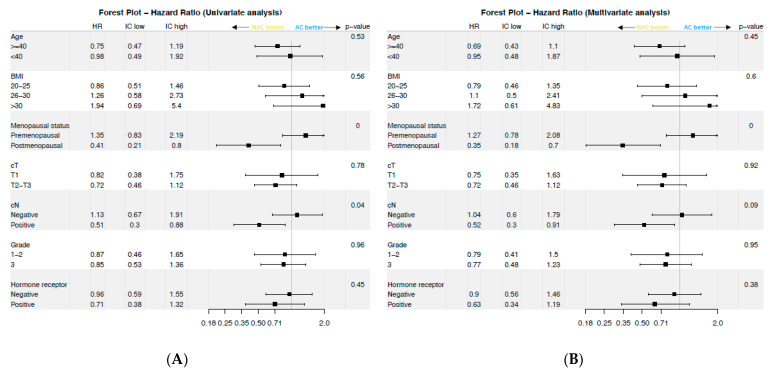
Impact of the strategy according to different clinical/histological variables. (**A**) Univariable analysis (**B**) Multivariable analysis. cN: initial clinical nodal status; cT: initial clinical tumor T stage; NAC: Neoadjuvant chemotherapy; AC: Adjuvant chemotherapy; BMI: body mass index; HR Hazard ratio; CI: confidence interval).

**Table 1 cancers-13-00370-t001:** Patient characteristics, by chemotherapy strategy (NAC vs. AC).

Variables	Category	AC (*n* = 701)	NAC (*n* = 202)	*p*-Value
**Age**		53.6 (25.4–85.5)	48.2 (26.7–79.4)	<0.0001
**BMI** (kg/m^2^)		23.4 ((15.6–50)	23.4 (17.2–43)	0.19
**Menopausal status**	Menopausal	397 (56.6)	80 (39.4)	<0.0001
	No menopausal	304 (43.4)	122 (60.6)	
**Grade**	1	14 (2)	3 (1.5)	<0.001
	2	231 (33)	1 (0.5)	
	3	456 (65)	199 (98	
**HR**	Negative	317 (45.2)	91 (44.8)	0.92
	Positive	384 (54.8)	111 (55.2)	
**Nodal status**	N0	573 (81.7)	81 (39.9)	<0.0001
	N1	123 (17.5)	111 (54.7)	
	N2	3 (0.4)	7 (3.4)	
	N3	2 (0.3)	4 (2)	
**Tumor size (mm)**		20 (3–100)	40 (10–120)	<0.0001
**T stage**	T1	453 (64.6)	24 (11.8)	<0.0001
	T2–T3	248 (35.4)	179 (88.2)	
**Surgery**	Lumpectomy	287 (40.9)	143 (70.4)	<0.0001
	Mastectomy	414 (59.1)	60 (29.6)	
**Axillary surgery**	Sentinel node biopsy	182 (26)	9 (4.4)	<0.0001
	Axillary dissection	519 (74)	194 (95.6)	
**Radiotherapy**		689 (98.3)	202 (99.5)	0.32
**Chemotherapy**	Anthra regimen	101 (14)	3 (1)	<0.001
	Tax regimen	12 (2)	16 (8)	
	Sequential Anthra-Tax regimen	546 (78)	175 (87)	
	Other	42 (6)	8 (4)	
**Endocrine therapy**		362 (51.6)	100 (49.5)	0.9
**pCR**		-	96 (38.5)	
**ypN status**	ypN0	-	106 (52.5)	
	ypN positive	-	96 (47.5)	
**Median follow up (months)**		48.7	39.2	<0.01
Events	Local/loco-regional recurrence	18 (2.6)	11 (5.4)	
	Distant metastasis	40 (5.7)	14 (6.9)	
	Death	18 (2.6)	2 (1)	

Qualitative variables are presented as the median (range); quantitative variables are presented as *n* (%); AC: Adjuvant chemotherapy; NAC: neoadjuvant chemotherapy, BMI: body mass index; HR: hormone receptor; pCR: pathological complete response; anthra: anthracyclines, Tax: taxane.

## Data Availability

Data available on request due to privacy/ethical restrictions. The data that support the findings of this study are available on request from the corresponding author, [FR]. The data are not publicly available because they contain information that could compromise the privacy of research participants.

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
