# Peer review of "HER2-Positive Breast Cancer Patients with Pre-Treatment Axillary Involvement or Postmenopausal Status Benefit from Neoadjuvant Rather than Adjuvant Chemotherapy Plus Trastuzumab Regimens"

_cancers, 2021, doi:10.3390/cancers13030370_

Round 1

Reviewer 1 Report

This is an important retrospective study providing relevant information for the clinical management of breast cancer patients. The study and analyses are properly designed and the paper well focused and clearly written.

The conclusion that NAC and AC differently behaves in specific subgroups of breast cancer patients (i.e. cN positive postmenopausal patients) is sound and behaves relevance to develop more personalized approaches in cancer care.

Given the relevance of the examined cohort, some additional analyses and comments could provide interesting information, as below specified:

Results. The prevalence of HER positive cancer (55%) is higher than those reported by other studies (20%) (see e.g. ref1). This finding should be explained and commented.

Results. Was BRCA1/2 mutation analysis performed? How can it be excluded that a subgroup of 5-10% BRCA1/2 positive patients could bias the presented results? Indeed NAC and CA could behave differently in BRCA positive and negative patients. This data is important especially in young patients that are included in the two cohorts (age <30 yo).

Results. There were difference between young and old patients? It would be of interest comparing NAC and AC effects in younger versus older patients (i.e. this included in the lowest or highest quartile of the age distribution). Indeed, age affects clinical outcomes of breast cancer therapies.

Results, lines 188-189. The statement that’ no other factor was associated with differences in CT strategies’ should be examine more accurately. No data is reported dealing the difference between high and low risk patients. It would be of interest to differentiate patients in high (early menarche age, nulliparity, late age at first pregnancy) as compared to low risk (devoid or risk factors). Are there differences in the outcome of CT, NAC, and AC between these two groups?

Minor point:

Abstract, line 24. Delete the word ‘significant’ because the P value is above the standard statistical significance threshold of P<0.05 (P=0.08). However, giving the CI values, it is clear that the interaction exists. Furthermore, in the text (line 139), but not in the abstract, it is reported that P threshold adopted is P<0.10 due to the lack of statistical power.

Reviewer 2 Report

Laas et al. present a retrospective analysis of patients with HER2-positive breast cancer, either receiving neoadjuvant chemotherapy (NAC) plus trastuzumab (n = 202) or adjuvant chemotherapy (AC) plus trastuzumab (n = 701). The authors state in the introduction that NAC is the current standard for patients with early breast cancer including those patients with HER2-positive tumors with no difference in survival reported so far, however, that the HER2 status was not routinely evaluated in the cited equivalence studies. The main endpoints of the comparative analysis of these two cohorts are disease-free survival (DFS) and overall survival (OS). Inverse probability of treatment weights (IPTW) were used to adjust imbalances between the patient cohorts. 

The results of this retrospective study provide additional evidence about the benefit of NAC in combination with trastuzumab for HER2-positive early breast cancer. Due to the retrospective design of the study, the study cohorts were rather different in size (202 vs. 701 patients), limiting their comparability.

Major comments

In the abstract, the authors point out that all patients received trastuzumab, however, there is no further description in the manuscript about the dosage or route of administration for all patients, except that trastuzumab was administered weekly after the end of chemotherapy. In case national guidelines are mentioned, the exact version along with a reference should be provided as current guidelines might be different from previous versions for 2005–2012. The authors should use trastuzumab throughout the manuscript as other phrases ‘’Overall, 903 patients with HER2-positive BC …. with HER2-targeted therapy (Table 1).’’ (lines 142+143) might be confusing. Also include references to European guidelines (line 82) and ASCO recommendations (line 85).

As pointed out in the discussion, one limitation of the study was that none of the patients received pertuzumab. Did none of the HER2-positive patients treated at the Institut Curie from 2005–2012 receive pertuzumab? If no patients received pertuzumab, the reasons for this choice should be pointed out; if patients received pertuzumab (or any other anti-HER2 targeted therapy) reasons for excluding patients from this analysis should be illustrated. If no patients received pertuzumab due to it being off-label for NAC during the period of analysis, it should be clearly stated.  

Laas et al. refer to the EBCTCG meta-analysis (lines 52-56) evaluating the impact of NAC versus AC treatment strategies on OS, distant recurrence and breast cancer mortality which all did not show significant differences between treatment groups. However, the meta-analysis showed that NAC was associated with more frequent local recurrence than AC (https://www.thelancet.com/journals/lanonc/article/PIIS1470-2045(17)30777-5/fulltext). It would be interesting to know if the addition of trastuzumab to NAC and AC regimens resolved this difference. Therefore, additional data for local recurrence rate and distant disease free survival would be necessary.

As mentioned in the title and the manuscript, especially patients with clinically node-positive disease (accounting for 60.1% of the NAC cohort) prior to treatment experienced a benefit form NAC in combination with trastuzumab. There are no information in Table 1 about post-treatment nodal status in the NAC group. The term pCR was not defined, e.g. both yT0 and yN0. Regardless of positive or negative post-treatment status, 95.5% of the NAC + trastuzumab cohort underwent axillary dissection, which has nowadays been partially replaced by targeted procedures, e.g. targeted axillary dissection. The reviewer suggests including a couple sentences on this topic in the discussion part.

According to the authors, patients with postmenopausal status also benefit from NAC + trastuzumab. In Table 1, only “menopausal status” is listed with no indication of pre- and post-menopausal status.

Minor comments

A “Simple Summary” as part of journal requirements has not been included in the manuscript.

The use of IPTW as a tool for reduction of bias and imparting comparability to study cohorts needs to be bolstered by more recent citations for its use (e.g. PMID: 26238958, 25934643), especially for analysis of survival outcomes in breast cancer patients (PMID: 31615634, 32315295, 31919848).

It needs to be stated how many patients in each group received endocrine therapy (line 108) and implications, if at all. It also needs to be clarified whether antihormonal therapy was given only to patients with residual tumor.

It should also be emphasized in lines 276-280 once again that NAC should be a standard, since in patients with non-pCR, T-DM1 as a post-neoadjuvant therapy could show further improvement in DFS as observed in the Katherine study.

The imbalance in propensity scores and further rectification through IPTW is wrongly ascribed to Fig 2E and 2F (instead of 2D) in lines 174-176.

Should it be “clinical tumor size” instead of “clinical size” in line 75?

A short section on the strengths of these analyses and how they build up current evidence could be included.

There are several inconsistencies in formatting, quality of images, abbreviations, etc. Some of these are listed here:

  • “node-positive patients [26]” (correct) versus “HER2-positive tumors[29]” (not correct, space missing in between tumors and [29]).
  • 1 legend: “(A)Whole population; (B) : according to menopausal status”
  • Content in the first and second column of Table 1 looks rather scrambled. It might help to place the content left-sided.
  • The contents in Figure 2 are out of focus. In Fig. 4 there are several format errors (e.g. below BMI). The font size in Figs. 3 and 4 is considerably smaller than the font size used in the text.
  • Please provide an expansion of “IPTW” at its first use in the abstract. Also, abbreviations have been inconsistently provided for Tables and Figures.

The overall quality of English will need to be improved substantially in order to impart clear communication of the research findings and the key messages.
